# Associations between Emotional Distress, Sleep Changes, Decreased Tooth Brushing Frequency, Self-Reported Oral Ulcers and SARS-Cov-2 Infection during the First Wave of the COVID-19 Pandemic: A Global Survey

**DOI:** 10.3390/ijerph191811550

**Published:** 2022-09-14

**Authors:** Morenike Oluwatoyin Folayan, Roberto Ariel Abeldaño Zuniga, Oliver C. Ezechi, Brandon Brown, Annie L. Nguyen, Nourhan M. Aly, Passent Ellakany, Ifeoma E. Idigbe, Abeedha Tu-Allah Khan, Folake Barakat Lawal, Mohammed Jafer, Balgis Gaffar, Bamidele Olubukola Popoola, Mir Faeq Ali Quadri, Jorma I. Virtanen, Joanne Lusher, Maha El Tantawi

**Affiliations:** 1Mental Health and Wellness Study Group, Obafemi Awolowo University, Ile-Ife 220282, Nigeria; 2Department of Child Dental Health, Obafemi Awolowo University, Ile-Ife 220282, Nigeria; 3Post Graduate School, University of Sierra Sur, Oaxaca 70805, Mexico; 4Department of Clinical Sciences, Nigerian Institute of Medical Research, Lagos 100001, Nigeria; 5Department of Social Medicine, Population and Public Health, Riverside School of Medicine, University of California, Riverside, CA 92501, USA; 6Department of Family Medicine, Keck School of Medicine, University of Southern California, Los Angeles, CA 90089, USA; 7Department of Pediatric Dentistry and Dental Public Health, Faculty of Dentistry, Alexandria University, Alexandria 21544, Egypt; 8Department of Substitutive Dental Sciences, College of Dentistry, Imam Abdulrahman Bin Faisal University, Dammam 31441, Saudi Arabia; 9Clinical Sciences Department, Nigerian Institute of Medical Research, Lagos 100001, Nigeria; 10Department of Biological Sciences, Faculty of Science, Superior University Lahore, Main Raiwind Road Punjab, Lahore 54590, Pakistan; 11Department of Periodontology and Community Dentistry, University of Ibadan and University College Hospital, Ibadan 200132, Nigeria; 12Preventive Dental Sciences Department, College of Dentistry, Jazan University, Jazan 45142, Saudi Arabia; 13Department of Preventive Dental Sciences, College of Dentistry, Imam Abdulrahman bin Faisal University, Dammam 31441, Saudi Arabia; 14Department of Child Oral Health, University of Ibadan, Ibadan 200132, Nigeria; 15Department of Oral Health Sciences, School of Dentistry, University of Washington, Seattle, WA 98105, USA; 16Faculty of Medicine, University of Turku, 20014 Turku, Finland; 17Provost’s Group, Regent’s University London, London NW1 4NS, UK

**Keywords:** oral health, oral ulcers, sleep disorders, emotions, COVID-19 pandemic

## Abstract

This study assessed the association between emotional distress, sleep changes, decreased frequency of tooth brushing, and self-reported oral ulcers, and the association between COVID-19 status and decreased frequency of tooth brushing. Using a cross-sectional online survey, data were collected from adults in 152 countries between July and December 2020. Binary logistic regression analyses were conducted to determine the associations between dependent (decreased frequency of tooth brushing, oral ulcers, change in sleep pattern) and independent (tested positive for COVID-19, depression, anxiety, frustration/boredom, loneliness, anger, and grief/feeling of loss) variables after adjusting for confounders (age, sex, level of education, employment status). Of the 14,970 participants data analyzed, 1856 (12.4%) tested positive for COVID-19. Respondents who reported feeling depressed (AoR: 1.375), lonely (AoR: 1.185), angry (AoR: 1.299), and experienced sleep changes (AoR:1.466) had significantly higher odds of decreased tooth brushing frequency. Respondents who felt anxious (AoR: 1.255), angry (AoR: 1.510), grief/sense of loss (AoR: 1.236), and sleep changes (AoR: 1.262) had significantly higher odds of oral ulcers. Respondents who tested positive for COVID-19 had significantly higher odds of decreased tooth brushing frequency (AoR: 1.237) and oral ulcers (AoR: 2.780). These findings highlight that the relationship between emotional distress and oral health may intensify during a pandemic.

## 1. Introduction

Poor oral health is a risk factor for general health [1]. A significant risk factor for systemic diseases is dental plaque. Plaque accumulation induces the development of periodontal diseases and the dissemination of bacteria into the bloodstream. The dissemination of bacteria results in pro-inflammatory events by atheroma formation, maturation, and exacerbation [2,3]. Periodontitis (inflammation of the soft tissue and the bone supporting the tooth) is a risk factor for several major illnesses, including type 2 diabetes, coronary artery disease, cerebral vascular disease, rheumatoid arthritis, chronic obstructive pulmonary disease, pneumonia, chronic kidney disease, cognitive impairment, obesity, metabolic syndrome, and cancer [4,5,6]. Periodontitis can also be non-dental biofilm induced [7].

Frequent tooth brushing effectively reduces the accumulation of plaque and calculus in humans [8]. Poor tooth brushing and plaque accumulation are also risk factors for dental caries [9,10]. Like periodontal diseases, dental caries can lead to the dissemination of bacteria that increase the risk for impaired growth, cardiovascular disease, head and neck cancer, and diseases of the immune system and kidney [11].

The frequency of tooth brushing is lower in people dealing with psychological distress [12], such as depression [13] and suicidal ideation [14], and psychological well-being during a crisis like the COVID-19 pandemic is associated with a decrease in tooth brushing frequency [15]. Psychological distress can be caused by emotional stress like loneliness [16], dental anxiety [17], frustration or boredom [18], anger [19], grief, or feeling of loss [20]. These factors are associated with oral diseases. 

Emotions regulate human behavior and moderate responses to stressful situations [21,22,23]. This, in turn, can affect oral health outcomes [24]. Sleep regulates emotions and the quality and amount of sleep influence how an individual reacts to events, thereby affecting general well-being [25]. Sleep problems have been featured commonly during the COVID-19 pandemic and are associated with higher levels of psychological distress [26]. Oral health and sleep quality are intrinsically linked [27]. However, little is known about how sleep quantity is associated with tooth brushing habits. 

Nevertheless, psychological distress and sleep disorders are risk factors for oral ulcers [28,29]. COVID-19-induced psychological distress and sleep disorders may be associated with recurrent aphthous ulcers due to elevated levels of salivary cortisol or reactive oxygen species in the saliva [28]. One study reported associations between psychological distress, sleep disorders experienced during the COVID-19 pandemic, reduced tooth brushing frequency, and the presence of oral ulcers [30]. 

Also, SARS-CoV-2 infects human cells using the ACE2 receptors found in the upper respiratory tract and the epithelial cells lining the salivary gland ducts. The virus can also be found on the dorsum of the tongue, and tooth brushing, along with other oral hygiene methods, may reduce the viral load in the oral cavity [31,32,33,34,35]. However, little is known about this plausible association between tooth brushing and COVID-19. 

This study tried to address some of these gaps in understanding the relationship between oral health and COVID-19. We assessed the association between COVID-19-induced emotional distress, sleep changes, reduced tooth brushing frequency, and the presence of self-reported oral ulcers. The study also assessed the association between decreased tooth brushing frequency and testing positive for COVID-19.

## 2. Materials and Methods

This study is part of a large cross-sectional multi-country study that collected data from 152 countries between July and December 2020 using an online platform (Survey Monkey^®^) that generated a convenient sample of 21,206 adults. The survey was designed to determine the impact of COVID-19 on the mental health and wellness of adults using a questionnaire that was validated for global use [36]. The overall Content Validity Index of the questionnaire was 0.83.

### 2.1. Ethical Consideration

Ethical approval was obtained from the Human Research Ethics Committee at the Institute of Public Health of the Obafemi Awolowo University Ile-Ife, Nigeria (HREC No: IPHOAU/12/1557). Additional ethical approval was obtained from India (D-1791-uz and D-1790-uz), Saudi Arabia (CODJU-2006F), Brazil (CAAE N° 38423820.2.0000.0010), and the United Kingdom (13283/10570). Study participants provided consent before participating in the online survey.

### 2.2. Recruitment of Study Participants

Participants were recruited through respondent-driven sampling. The survey link, created using an online survey tool (Survey Monkey^®^), was posted on social media groups (Facebook, Twitter, and Instagram), network email lists, and WhatsApp groups. Participants were included if they were above 18 years of age, could understand the survey language, and could access the survey using an electronic device and an internet connection. Respondents were required to provide details about their sociodemographic profile, medical health profile, oral health status, pandemic stress level, and the impact of COVID-19 on their daily life. 

### 2.3. Study Procedure

The survey was preceded by a brief introduction explaining the purpose of the study, assuring participants of their voluntary participation and confidentiality of their data. Before proceeding, participants were required to check a box that indicated consent. The questionnaire took, on average, 11 min to complete and was administered in English. Multiple best-practice procedures were performed to increase the data quality of the survey [37,38]. Each participant could only complete a single questionnaire through IP address restrictions, though they could edit their answers freely until they chose to submit. Full details of the methodology can be found elsewhere [39,40,41].

### 2.4. Control Variables

#### Sociodemographic Variables

The confounding sociodemographic variables were age in years, sex at birth (male, female, others), the highest level of education attained (none, primary, secondary, college/university), and employment status (employed, unemployed, student, retiree).

### 2.5. Dependent Variables

#### Oral Health

Two indicators of oral health status were determined using two questions: did the frequency of tooth brushing change during the lockdown with the options of ‘Yes (increased)’, ‘Yes (decreased)’ and ‘No change’; and did you have oral ulcers during the lockdown with the option responses of ‘yes’ or ‘no’. 

### 2.6. Independent Variables

#### 2.6.1. COVID-19 Status

This was determined by asking respondents if they had tested positive for COVID-19. The response options were ‘yes’ or ‘no’. 

#### 2.6.2. Emotional Distress

Respondents were asked to indicate if they had experienced any of the ten listed emotions during the pandemic. The list included depression, anxiety, frustration or boredom, loneliness, anger, grief, or feeling of loss. Respondents were required to tick a checkbox against any emotions they experienced during the pandemic. 

#### 2.6.3. Sleep Changes

Respondents were asked to indicate if they had experienced a change (sleeping more, sleeping less or other changes from normal) or no change in sleep pattern. Each respondent was required to check a response box that indicated if they had experienced a change in sleep pattern during the pandemic. For this study’s statistical analysis, the responses were dichotomized into sleeping changes present (sleeping more, sleeping less, other changes from normal) and sleeping disorders absent (no change in the sleeping pattern).

### 2.7. Data Analyses

Raw data were downloaded, cleaned, and imported to SPSS version 23.0 (IBM Corp., Armonk, NY, USA) for analysis. Where appropriate, chi-square tests and t-tests were used to assess the associations between the dependent, independent, and confounding variables. Inferential analyses were conducted using binary logistic regression analysis models and adjusting for confounding variables (sociodemographic variables). Adjusted odds ratios (AoR) for the binary logistic regression model and 95% confidence intervals (CI) were calculated. Statistical significance was set at <0.05.

## 3. Results

As shown in Table 1, 14,970 participants provided complete data for analysis. The mean (standard deviation) age of the participants was 34.58 (12.7) years, 9328 (62.3%) were females, 1636 (10.9%) reported a decrease in tooth brushing frequency during the lockdown, and 1760 (11.8%) reported having oral ulcers. Also, 1856 (12.4%) tested positive for COVID-19, 4394 (29.4%) felt frustrated or bored, 2416 (16.1%) were depressed, 4262 (28.5%) anxious, 2760 (18.4%) lonely, 1841 (12.3%) felt angry; and 1546 (10.3%) felt grief or had a feeling of loss. Finally, 4715 (31.5%) reported sleep changes.

Significantly more respondents who were younger (*p* < 0.001), students (*p* < 0.001), felt frustrated or bored (*p* < 0.001), anxious (*p* < 0.001), depressed (*p* < 0.001), lonely (*p* < 0.001), angry (*p* < 0.001), grief or sense of loss (*p* < 0.001), had sleep changes (*p* < 0.001) and tested positive for COVID-19 (*p* < 0.001) reported a decrease in the frequency of tooth brushing during the lockdown. Also, significantly fewer people with no formal education (*p* < 0.001) reported decreased tooth brushing frequency during the lockdown.

Similarly, significantly more respondents who were younger (*p* < 0.001), had primary school education (*p* < 0.001), were students (*p* < 0.001), felt frustrated or bored (*p* < 0.001), anxious (*p* < 0.001), depressed (*p* < 0.001), lonely (*p* < 0.001), angry (*p* < 0.001), grief or sense of loss (*p* < 0.001), had sleep changes (*p* < 0.001) and tested positive for COVID-19 (*p* < 0.001) reported having oral ulcers during the lockdown. Also, significantly fewer females (*p* < 0.001) and people who had no formal education (*p* < 0.001) reported having oral ulcers during the lockdown.

Table 2 shows that respondents with primary school level of education (AoR: 1.414; 95% CI: 1.034–1.934; *p* = 0.030), students (AoR:1.449; 95% CI: 1.217–1.724; *p* < 0.001), those who reported feeling depressed (AoR:1.375; 95% CI: 1.185–1.596; *p* < 0.001), lonely (AoR: 1.185; 95% CI: 1.025–1.369; *p* = 0.022), angry (AoR:1.299; 95% CI: 1.107–1.525; *p* = 0.001), had sleep changes (AoR: 1.466; 95% CI: 1.303–1.650; *p* < 0.001) and tested positive for COVID-19 (AoR:1.237; 95% CI: 1.069–1.433; *p* = 0.004), had higher odds of reporting a decrease in tooth brushing frequency during the pandemic. On the contrary, older respondents (AoR:0.984; 95% CI: 0.978–0.990; *p* < 0.001) had lower odds of reporting a decrease in tooth brushing frequency during the pandemic when there was a lockdown.

Respondents with primary school level of education (AoR: 2.619; 95% CI: 2.011–3.411; *p* < 0.001)), those employed (AoR: 1.441; 95% CI: 1.221–1.701; *p* < 0.001), retired (A0R: 1.876; 95% CI: 1.305–2.697; *p* = 0.001), those who reported feeling anxious (AoR: 1.255; 95% CI: 1.106–1.423; *p* < 0.001), angry (AoR: 1.510; 95% CI: 1.293–1.763; *p* < 0.001), grief or a sense of loss (AoR: 1.236; 95% CI: 1.047–1.460; *p* = 0.013), had sleep changes (AoR: 1.262; 95% CI: 1.122–1.419; *p* < 0.001) and tested positive for COVID-19 (AoR: 2.780; 95% CI: 2.460–3.141; *p* < 0.001), had higher odds of reporting oral ulcers during the pandemic. On the contrary, older respondents (AoR: 0.977; 95% CI: 0.971–0.982; *p* < 0.001) had lower odds of decreased tooth brushing during the pandemic

## 4. Discussion

According to the present findings, COVID-19-induced emotional distress and sleep changes were associated with decreased frequency of tooth brushing and an increased risk for oral ulcers during the lockdown. The types of emotional distress associated with a decrease in the frequency of tooth brushing were not always associated with a higher risk of oral ulcers. While depression, loneliness, and anger were associated with a decrease in the frequency of tooth brushing, anxiety, anger, and grief or sense of loss were associated with a higher risk of oral ulcers. There was also a significant association between frequency of tooth brushing, presence of oral ulcers, and testing positive for COVID-19.

This study provides a global perspective on the likely impact of COVID-19 lockdowns on oral health and the possible impact of the pandemic on oral and systemic health-related behaviors. The study, however, had a few limitations. It was a cross-sectional study, and data were skewed to those with higher educational levels due to the non-probability sampling technique conducted using an online platform that inadvertently excluded populations less able to access or use online services [42]. Non-probability sampling techniques were appropriate during the pandemic when reduced physical contact and human movement regulations were in place. The large sample size compensates for some of the limitations of the non-probability sampling by allowing for a more precise estimate of effects, which provides a reduced margin for error and an increase in the generalizability of the study findings due to the cosmopolitan nature of this study [43,44]. 

The findings suggest that depression, loneliness, and anger may result in negative lifestyle modifications associated with a lack of willpower [45,46]. Depression may lead to adopting poor health behaviors; consequently, in this case, a decrease in tooth brushing. Loneliness, described as the subjective experience of social isolation, is also associated with adopting adverse health behaviors such as poor diet, physical inactivity, and problematic smoking and alcohol intake [47,48,49,50,51,52,53]. Anger is also associated with poor health behaviors [54]. The present study provides first-time evidence that a negative oral health behavior associated with the pandemic, decreased frequency of tooth brushing, may be associated with loneliness and anger.

On the other hand, anxiety, anger, grief, or a sense of loss may result in disorders of the body’s physiology that may lead to oral ulcers. Suggested disorders associated with anxiety include a transitory rise in salivary cortisol and reactive oxygen species in the saliva [55]. In the same context, grief is associated with dysregulation of the immune function resulting in the elevation in inflammation-related health problems [56,57,58,59,60,61,62], one of which is recurrent oral ulcers [63,64]. 

The current study’s findings implicate anger in both behavior and physiobiological changes. A prior study indicated that anger reprioritizes neural and physiological resources that negatively affect behavior and physiological changes. That is, anger decreases liking judgments and enhances avoidance behaviors, negatively affecting health behavior. [65,66]. Anger may have moderated the association between sleep changes, decreased tooth brushing frequency, and increased risk of oral ulcers during the lockdown. Sleep changes are associated with poor health outcomes [67,68], causing functional deficits, increased cortisol secretion, and abnormal growth hormone metabolism [69]. Anger may have also caused oral ulcers through the same physiological process as sleep. However, the moderating effect of anger on the association between sleep changes and decreased tooth brushing may be less direct. Sleep deprivation is associated with anger, irritability, aggression, and short temper, as well as mood swings, leading to decreased motivation and positive health behavior [70]. These are postulations that need to be studied further.

The findings of this current study also highlighted a negative effect of COVID-19-associated emotional distress and sleep changes on oral health with respect to tooth brushing frequency and oral ulcers. Prior studies indicated that aphthous, hemorrhagic, and necrotic oral ulcers were associated with COVID-19 [71,72]. Previous studies also suggested that the pandemic is associated with a higher risk of reduced tooth brushing frequency [15,73,74]. This raises concern over the long-term impact of the pandemic on people at risk of periodontal-induced health risks [7]. The finding of this study suggests possible pathways for the observed results: COVID-19-induced emotional distress and sleep changes may be linked to a decrease in the frequency of tooth brushing and an increased risk for oral ulcers during the lockdown. These study findings suggest the need for reinforcing information about the importance of tooth brushing during pandemic situations and the necessity to support individuals with emotional distress with oral hygiene practice. Such support would assist in preventing physiological detriment caused by poor oral health during a pandemic. 

In addition, the current study suggests a possible link between a reduction in the frequency of tooth brushing and testing positive for COVID-19. There are speculations on this association between tooth brushing andSARS-CoV-2 infection risk, with oral hygiene practices reducing this risk [75]. Though the current study is unable to determine the direction of this association, the present finding suggests a plausibility that needs to be explored further.

## 5. Conclusions

In conclusion, COVID-19-associated emotional distress and disrupted sleep patterns were associated with a reduction in tooth brushing frequency and the presence of oral ulcers, possibly through physiobiological and behavioral pathways. It would be worthwhile for further studies to substantiate this initial observation of plausibility that decreased tooth brushing frequency may be associated with an increased risk of SARS-CoV-2 infection. 

## Figures and Tables

**Table 1 ijerph-19-11550-t001:** Factors associated with decreased tooth brushing frequency and self-reported oral ulcers during the lockdown in the first wave of the COVID-19 pandemic in 152 countries (N = 14,970).

Variables	Total N = 14,970	Tooth Brushing Decreased during the Lockdown		Chi-Square Test *p* Value	Self-Reported Oral Ulcers during the Lockdown		Chi-Square Test *p* Value
		Yes	No		Yes	No	
		N = 1636	N = 13,334		N = 1760	N = 13,210	
		10.9%	89.1%		11.8%	88.2%	
		n (%)	n (%)		n (%)	n (%)	
** Age	34.58 (SD 12.7)	31.20 (SD 12.1)	34.99 (SD 12.8)	* < 0.001	32.15 (SD 11.35)	34.90 (SD 12.9)	* < 0.001
Sex at birth							
Male	5544 (37.0)	593 (10.7)	4951 (89.3)		739 (13.3)	4805 (86.7)	
Female	9328 (62.3)	1025 (11.0)	8303 (89.0)	5.913	1004 (10.8)	8324 (89.2)	* 25.048
Others	98 (0.7)	18 (18.4)	80 (81.6)	0.052	17 (17.3)	81 (82.7)	<0.001
Educational level							
No formal education	305 (2.0)	22 (7.2)	283 (92.8)		33 (10.8)	272 (89.2)	
Primary	376 (2.5)	49 (13.0)	327 (87.0)	* 22.496	81 (21.5)	295 (78.5)	* 36.367
Secondary	2709 (18.1)	356 (13.1)	2353 (86.9)	<0.001	324 (12.0)	2385 (88.0)	<0.001
College/university	11,580 (77.4)	1209 (10.4)	10,371 (89.6)		1322 (11.4)	10,258 (88.6)	
Employment status							
Employed	8305 (55.5)	736 (8.9)	7569 (91.1)		990 (11.9)	7315 (88.1)	
Unemployed	3736 (25.0)	238 (10.1)	2113 (89.9)	* 163.641	217 (9.2)	2134 (90.8)	* 27.938
Student	578 (3.9)	617 (16.5)	3119 (83.5)	<0.001	500 (13.4)	3236 (86.6)	<0.001
Retiree	2351 (15.7)	45 (7.8)	533 (92.2)		53 (9.2)	525 (90.8)	
Frustration or boredom							
Yes	4394 (29.4)	596 (13.6)	3798 (86.4)	* 44.377	611 (13.9)	3783 (86.1)	* 27.673
No	10,576 (70.6)	1040 (9.8)	9536 (90.2)	<0.001	1149 (10.9)	9427 (89.1)	<0.001
Anxiety							
Yes	4262 (28.5)	579 (13.6)	3683 (86.4)	* 43.201	656 (15.4)	3606 (84.6)	* 75.886
No	10,708 (71.5)	1057 (9.9)	9651 (90.1)	<0.001	1104 (10.3)	9604 (89.7)	<0.001
Depression							
Yes	2416 (16.1)	408 (16.9)	2008 (83.1)	* 105.092	388 (16.1)	2028 (83.9)	* 54.411
No	12,554 (83.9)	1228 (9.8)	11,326 (90.2)	<0.001	1372 (10.9)	11,182 (89.7)	<0.001
Loneliness							
Yes	2760 (18.4)	435 (15.8)	2325 (84.2)	* 81.177	417 (15.1)	2343 (84.9)	* 36.645
No	12,210 (81.6)	1201 (9.8)	11,009 (90.2)	<0.001	1343 (11.0)	10,867 (89.0)	<0.001
Anger							
Yes	1841 (12.3)	312 (16.9)	1529 (83.1)	* 78.120	351 (19.1)	1490 (80.9)	* 108.087
No	13,129 (87.7)	1324 (10.1)	11,805 (89.9)	<0.001	1409 (10.7)	11,720 (89.3)	<0.001
Grief or feelings of loss							
Yes	1546 (10.3)	229 (14.8)	1317 (85.2)	* 26.717	279 (18.0)	1267 (82.0)	* 65.742
No	13,424 (89.7)	1407 (10.5)	12,017 (89.5)	<0.001	1481 (11.0)	11,943 (89.0)	<0.001
Sleep changes							
Yes	4715 (31.5)	704 (14.9)	4011 (85.1)	* 113.277	695 (14.7)	4020 (85.3)	* 59.048
No	10,255 (68.5)	932 (9.1)	9323 (90.9)	<0.001	1065 (10.4)	9190 (89.6)	<0.001
Tested positive to COVID-19							
Yes	1856 (12.4)	256 (13.8)	1600 (86.2)	* 17.860	470 (25.3)	1386 (74.7)	* 375.858
No	13,114 (87.6)	1380 (10.5)	11,734 (89.5)	<0.001	1290 (9.8)	11,824 (90.2)	<0.001

** *t*-test conducted, SD: Standard Deviation, * statistically significant.

**Table 2 ijerph-19-11550-t002:** The binary logistic regression analysis determines the risk indicators for the decreased frequency of tooth brushing and the presence of self-reported oral ulcers during the first wave of COVID-19 in 152 countries (N = 14,970).

Variables	Decreased Tooth Brushing Frequency	Self-Reported Oral Ulcers Present
	AoR; 95% CI (*p* Values)	AoR; 95% CI (*p* Values)
Age	* 0.984; 0.978–0.990 (*p* < 0.001)	* 0.977; 0.971–0.982 (*p* < 0.001)
Sex at birth		
Male	0.658; 0.386–1.121; (*p* = 0.124)	0.880; 0.506–1.532 (*p* = 0.651)
Female	0.599; 0.353–1.018; *p* = 0.058)	0.674; 0.388–1.171 (*p* = 0.162)
Others	1.000	1.000
Educational level		
No formal education	0.765; 0.489–1.198, (*p* = 0.242)	1.268; 0.863–1.864 (*p* = 0.226)
Primary	* 1.414; 1.034–1.934, (*p* = 0.030)	* 2.619; 2.011–3.411 (*p* < 0.001)
Secondary	1.029; 0.900–1.177, (*p* = 0.672)	1.006; 0.876–1.156 (*p* = 0.930)
College/university	1.000	1.000
Employment status		
Employed	0.965; 0.818–1.135 (*p* = 0.656)	* 1.441;1.221–1.701 (*p* < 0.001)
Student	* 1.449; 1.217–1.724 (*p* < 0.001)	1.199; 0.998–1.442 (*p* = 0.053)
Retiree	1.225; 0.839–1.789 (*p* = 0.294)	* 1.876; 1.305–2.697 (*p* = 0.001)
Unemployed	1.000	1.000
Frustration or boredom		
Yes	0.959; 0.843–1.090 (*p* = 0.522)	0.918; 0.809–1.042 (*p* = 0.187)
No	1.000	1.000
Anxiety		
Yes	1.081; 0.948–1.233 (*p* = 0.246)	* 1.255; 1.106–1.423 (*p* < 0.001)
No	1.000	1.000
Depression		
Yes	* 1.375; 1.185–1.596 (*p* < 0.001)	1.015; 0.873–1.180 (*p* = 0.848)
No	1.000	1.000
Loneliness		
Yes	* 1.185; 1.025–1.369 (*p* = 0.022)	0.970; 0.839–1.122 (*p* = 0.684)
No	1.000	1.000
Anger		
Yes	* 1.299; 1.107–1.525 (*p* = 0.001)	* 1.510; 1.293–1.763 (*p* < 0.001)
No	1.000	1.000
Grief or feelings of loss		
Yes	0.922; 0.773–1.099 (*p* = 0.362)	* 1.236; 1.047–1.460 (*p* = 0.013)
No	1.000	1.000
Sleep changes		
Yes	* 1.466; 1.303–1.650 (*p* < 0.001)	* 1.262; 1.122–1.419 (*p* < 0.001)
No	1.000	1.000
Tested positive for COVID-19		
Yes	* 1.237; 1.069–1.433 (*p* = 0.004)	* 2.780; 2.460–3.141 (*p* < 0.001)
No	1.000	1.000

* Statistically significant. AoR: Adjusted Odds Ratio, CI: Confidence Interval.

## Data Availability

The data presented in this study are available on request from the corresponding author. The data are not publicly available due to ongoing data analysis from the dataset.

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
