# Peer review of "Associations between Emotional Distress, Sleep Changes, Decreased Tooth Brushing Frequency, Self-Reported Oral Ulcers and SARS-Cov-2 Infection during the First Wave of the COVID-19 Pandemic: A Global Survey"

_ijerph, 2022, doi:10.3390/ijerph191811550_

Round 1

Reviewer 1 Report

Front page:

- before the title please remove the words “Type of the paper” and keep only “Article”

- edit the citation accordingly to the template 

Introduction

- in line 60 remove the ”s” between the words ”diseases” and ”and”

- in line 100: there are missing words in the following sentence: ”Therefore, the present study.”. Please remove it or complete it. 

- I would suggest adding a paragraph at the end of Introduction to specify the aim of the study with all the objectives that were mentioned at the end of each paragraphs in the introduction section. 

Material and method

- please rephrase ”formed part”

- please specify not only the number of the countries but also the total number of participants

- please specify how the Ethical Approval was valid or obtained for the other countries involved in the study

- in ”Recruitment of the participants” in the final sentence regarding the detailes required from the participants there is nothing mentioned about  the assesssed oral health-related aspects 

- in ”Data analysis” please modify ”statistical significance”: replace ”5%” with ”0.05”.

Results

- in line 178 please add ”%” after the ”18.4” in the brackets

Table 1

- in Table 1 in the top area please replace”x2” with ”Chi-square test” or the correct symbol

- in Table 1 when comparing the mean age (which is a numeric variable) between groups (Toothbruhing decrease Yes vs No; oral ulcer Yes vs No) the proper statistical test is the t-test. In Table 1 there is a column marking the Chi-square test, while the t-test is not mentioned for which variable was used. Please modify accordingly

- in Table 1 in this given form of the table, the values of Chi-square only show that there are differences in distribution of answers (Yes/No) between the subgroups in each variable (for example between College/university, Secondary, Primary and None for the ”Education level” variable), but without specifically showing that the subgroup ”None” had statistically signifficant less decrease in the frequency of toothbrushig compared to the others subgroups, as it is specified in the text: ”significantly less people who had no education (p<0.001) reported a decrease in their frequency of toothbrushing during the lockdown. ”. There are necessary more statistical tests to show which specific subgroup in each variable differs from the others. I recommend additional proper statistical tests and then add the results in the table accordingly.

Tabel 2

- in Tabe 2 for the AOR regarding the sex of the subjects, taking into consideration that the proportion of patients declaring their sex as “Other” I would suggests not using this as reference  (AOR=1). I consider more relevant in this particular case to calculate the Odd Ration on male related to female (Reference female = 1) or female related to male (Reference male = 1)

- In Tables 1 and 2 I suggest marking the results that are statistically signifficant (by adding the “ * “ symbol and add its meaning to the legend in the bottom of the tables

Line 212: add “.” at the end of the sentence. 

Discussion

Line 223: please replace “ ; ” with “ , “ in order for the sentence to make sence

Line 263: I suggest replacing “vis-a-vis” with an English synonym (for example “related to” or “regarding”)

Line 271: please consider finding a clearer expression for “oral ill-health”

Line 274: please insert space between “and” and “SARS-CoV-2”

Conclusion

- the sentence between 281 and 283: I consider hazardous such a conclusion, I do not agree that the results of the study support the affirmation even if it is mentioned as “plausible”, there is not enough evidence for this type of study to support that conclusion. I suggest removing this conclusion. I recommend in the conclusion to point out the variables that statistically significant are associated with the decrease in toothbrushing frequency and the increase in the frequency of oral ulcers.

Funding

- please consider the proper editing. Chose one of the statements in brackets

References

- items 2 and 6 are identical. Please revise

- please consider the proper editing of all the items according to the instructions for authors required by the IJERPH journal

Author Response

We thank the reviewers for taking out time to thoroughly review the manuscript and raise these essential points. We have addressed the issues raised. Below are the point by point response to the suggested revisions

Reviewer 1

Front page:

- before the title please remove the words “Type of the paper” and keep only “Article”

RESPONSE: thanks for the observation. This has been done

- edit the citation accordingly to the template 

 RESPONSE: I have left this to the publication managers as there are details I am unable to fill in

Introduction

- in line 60 remove the ”s” between the words ”diseases” and ”and”

RESPONSE: Thanks for picking this. Suggested edit effected

- in line 100: there are missing words in the following sentence: ”Therefore, the present study.”. Please remove it or complete it.

RESPONSE: Thanks for picking this. Suggested edit effected

- I would suggest adding a paragraph at the end of Introduction to specify the aim of the study with all the objectives that were mentioned at the end of each paragraphs in the introduction section. 

 RESPONSE: This correction has been effected. There is now a new paragraph dedicated to the study objectives.

Material and method

- please rephrase ”formed part”

 RESPONSE: Thanks for the suggestion. We changed ‘formed’ to ‘is’

- please specify not only the number of the countries but also the total number of participants

RESPONSE: suggested edit effected. We have included information on the number of participants – 21106 – and removed this information from the result section.

- please specify how the Ethical Approval was valid or obtained for the other countries involved in the study

RESPONSE: First, this was an online data collection process that used the social media and so we could collect data from anyone online. We did not collect data through face-to-face engagement. We also did not recruit field workers for data collection except in Nigeria. The large number of countries involved with the study were recruited online and not through country specific recruitment. The Main ethics approval was from the host country for the study (Nigeria). Data was collected by snowballing.

- in ”Recruitment of the participants” in the final sentence regarding the detailes required from the participants there is nothing mentioned about  the assesssed oral health-related aspects 

RESPONSE: thanks a million for picking this up. We have included this in the section.

- in ”Data analysis” please modify ”statistical significance”: replace ”5%” with ”0.05”.

 RESPONSE: Suggested edit effected.

Results

- in line 178 please add ”%” after the ”18.4” in the brackets

RESPONSE: Thanks for picking this up. Correction effected

Table 1

- in Table 1 in the top area please replace”x2” with ”Chi-square test” or the correct symbol

RESPONSE: Suggested edit effected

- in Table 1 when comparing the mean age (which is a numeric variable) between groups (Toothbruhing decrease Yes vs No; oral ulcer Yes vs No) the proper statistical test is the t-test. In Table 1 there is a column marking the Chi-square test, while the t-test is not mentioned for which variable was used. Please modify accordingly

RESPONSE: we have addressed this by putting an asterisk against age and then noting as a footnote that the analysis for age was done using a t-test. Thanks for the guidance.

- in Table 1 in this given form of the table, the values of Chi-square only show that there are differences in distribution of answers (Yes/No) between the subgroups in each variable (for example between College/university, Secondary, Primary and None for the ”Education level” variable), but without specifically showing that the subgroup ”None” had statistically signifficant less decrease in the frequency of toothbrushig compared to the others subgroups, as it is specified in the text: ”significantly less people who had no education (p<0.001) reported a decrease in their frequency of toothbrushing during the lockdown. ”. There are necessary more statistical tests to show which specific subgroup in each variable differs from the others. I recommend additional proper statistical tests and then add the results in the table accordingly.

RESPONSE: We tried hard to understand the issue raised here. We have changed the word none to no formal education. We have also included the result of the no formal education with respect to the presence of ulcers. We cannot identify any missing variables not reported.

Tabel 2

- in Tabe 2 for the AOR regarding the sex of the subjects, taking into consideration that the proportion of patients declaring their sex as “Other” I would suggests not using this as reference  (AOR=1). I consider more relevant in this particular case to calculate the Odd Ration on male related to female (Reference female = 1) or female related to male (Reference male = 1)

 RESPONSE: This was a serious consideration to make. We however felt it was important to highlight the issues of a minority through this manuscript. The study finding would have enabled us discuss extensively about a minority group that may otherwise be invisible. We have taken this path for all the manuscript published from this database going forward. We hope the reviewer will allow us do this as we hope, at the end of the series, to pull together all findings and highlight  the needs of sexual minorities during the pandemic.

- In Tables 1 and 2 I suggest marking the results that are statistically signifficant (by adding the “ * “ symbol and add its meaning to the legend in the bottom of the tables

RESPONSE: done

Line 212: add “.” at the end of the sentence. 

 RESPONSE: We searched and could not identify what was being referred too here

Discussion

Line 223: please replace “ ; ” with “ , “ in order for the sentence to make sence

RESPONSE: Thanks for picking this error up. Suggested edit effected.

Line 263: I suggest replacing “vis-a-vis” with an English synonym (for example “related to” or “regarding”)

RESPONSE: We have replaced the word with the phrase ‘with respect to’

Line 271: please consider finding a clearer expression for “oral ill-health”

RESPONSE: We have replaced this with poor oral health

Line 274: please insert space between “and” and “SARS-CoV-2”

RESPONSE:  Done. Thanks for spotting this.

Conclusion

- the sentence between 281 and 283: I consider hazardous such a conclusion, I do not agree that the results of the study support the affirmation even if it is mentioned as “plausible”, there is not enough evidence for this type of study to support that conclusion. I suggest removing this conclusion. I recommend in the conclusion to point out the variables that statistically significant are associated with the decrease in toothbrushing frequency and the increase in the frequency of oral ulcers.

RESPONSE: Thanks. We deleted this line. We rephrased the conclusion as: It would be worthwhile that further studies are conducted to substantiate this initial observation of a plausibility that decreased tooth brushing frequency may be associated with an increase in the risk of SARS-CoV-2 infection. 

Funding

- please consider the proper editing. Chose one of the statements in brackets

RESPONSE: Thanks for raising this. This has been edited

References

- items 2 and 6 are identical. Please revise

RESPONSE: Thanks for picking this up. We replaced reference 2 with Corredor Z, Suarez-Molina A, Fong, C. et al. Presence of periodontal pathogenic bacteria in blood of patients with coronary artery disease. Sci Rep. 2022; 12: 1241

- please consider the proper editing of all the items according to the instructions for authors required by the IJERPH journal

RESPONSE: we have reviewed the document again for edits. Thanks for the excellent support.

Reviewer 2 Report

Oral health and COVID 19 is a actual topic in social media.

Aphthous ulceration and disstress with worsening of selfperformed oral health care during a COVID 19-infection are of interest.

Author Response

  1. Change oral ulcers to aphthous ulcer

RESPONSE: The authors recognise that the stress associated with COVID-19 may most likely, lead to aphthous ulcer. We however did not ask about aphthous ulcers specifically. This has limited our ability to change any of the reference to oral ulcer to aphthous ulcers throughout the document

  1. Change the title to include a questionnaire survey on social media

RESPONSE: we appreciate the suggestion of the author. We however, feel this level of specificity may not be needed. We feel the title is good enough to reflect the information about the survey

  1. Location and centre of the groups

RESPONSE: This information has been included in the name of the group. We wrote: Obafemi Awolowo University, Ile-Ife, Nigeria

  1. Subjective association for people with COVID-19 infection

RESPONSE: We assume the reviewer will want the authors to include this information in the objective. We feel the objective is comprehensive without that level of detail. The methodology provides the details required by the reviewer.

  1. Number of participants with and without COVID-19 infection

RESPONSE: Thanks for this suggestion. We have included the number of the participants with COVID-19 in the information

  1. The risk factors for oral health are tobacco use and overweight. These were not included in the analysis

RESPONSE: the reviewer raised a significant point. We agree on this. However, these data were not included in this secondary data analysis plan. This is being raised for consideration in future data collection.

  1. Delete lines 53 and 54

RESPONSE: We agree on the need for this sentence to be deleted.

  1. New classification by Caton et al, 2018

RESPONSE: Thanks for this excellent reference. We have adopted it and used it to slightly modify the phrase

  1. Delete in animals since study did not include them

RESPONSE: done

  1. Please use reference on longitudinal study on plaque accumulation

RESPONSE: We have now replaced reference 8 with Löe H. Oral hygiene in the prevention of caries and periodontal disease. Int Dent J. 2000 Jun;50(3):129-39. doi: 10.1111/j.1875-595x.2000.tb00553.x

  1. Line 100-102 deleted as indicated. No health education model was tested. This has been deleted. A new paragraph was developed to pool the study objectives together as suggested by reviewer 1.
  2. Where did the study participants come from. There was no ethics approval from author countries

RESPONSE: this was an online data collection process that used the social media and so we could collect data from anyone online. We did not collect data through face-to-face engagement. We also did not recruit field workers for data collection except in Nigeria. The large number of countries involved with the study were recruited online and not through country specific recruitment. The main ethics approval was from the host country for the study (Nigeria). Data was collected by snowballing.

  1. Please enclose the questionnaire as an appendix

RESPONSE: The reference 36 is the reference to the validated questionnaire. The instrument is a supplementary file for that reference. The questionnaire can be accessed online through that reference

  1. The aetiology of aphthous ulcer and distress belongs to the introduction section

RESPONSE: Thanks for the suggestion. We note that the study examined oral ulcers and we were not specific about aphthous ulcer. We are therefore careful not to allude to our findings as aphthous ulcer. We think it is appropriate not to frontload the discussion with a focus on aphthous ulcer as this was not measured.

  1. Please explain what is new about your study that was not published by your first and corresponding author

RESPONSE: Thanks for raising the need to highlight new information generated from this study. We noted that: The finding of this study suggests possible pathways for the observed findings: COVID-19-induced emotional distress and sleep changes may be linked to a decrease in the frequency of tooth brushing and an increased risk for oral ulcers during the lockdown.

  1. The study was not a longitudinal study and so no conclusion on longitudinal effect can be made

RESPONSE: We agree with this observation and have deleted the statement. We concluded by noting: It would be worthwhile that further studies are conducted to substantiate this initial observation of a plausibility that decreased tooth brushing may lead to an increase in risk of SARS-CoV-2 infection. 

Round 2

Reviewer 1 Report

Thank you, the manuscript has been revised according to previous recommendations.

Author Response

Thanks a million for the support indeed

Reviewer 2 Report

Oral ulcers, by probands probably identified during Covid19 infection, can be a result of different bacterial, fungal and/or viral infections as well as a symptom of different bullous autoimmunological diseases or of injurys aetiology. The differentiation is very important. Allthought the authors did not differentiate oral ulcerations by further discreptions in their survey, the analysis on oral ulcers in COVID 19 in the paper and title should be resigned.

Author Response

Thanks a million for the comments. Like you rightly noted The reviewer editor 2 seem to require we named the oral ulcer aphthous ulcer. While this is likely, there are multiple literatures that indicates there are multiple possible ulcers that can be associated with the COVID-19. These include the virus induced ulceration. SARS-CoV-2 can be detected in saliva and oropharyngeal secretions and may infect the oral tissues and express its pathogenic mechanisms in the oral and oropharyngeal mucosae. COVID-19 associated oral lesions may present 2 patterns of oral ulcerations - aphthous-like and superficial necrosis—affecting multiple oral sites in patients diagnosed with COVID-19 (Brandão TB, Gueiros LA, Melo TS, Prado-Ribeiro AC, Nesrallah ACFA, Prado GVB, Santos-Silva AR, Migliorati CA. Oral lesions in patients with SARS-CoV-2 infection: could the oral cavity be a target organ? Oral Surg Oral Med Oral Pathol Oral Radiol. 2021 Feb;131(2):e45-e51. doi: 10.1016/j.oooo.2020.07.014)

The authors did not ask about the type of lesion. We only asked if participants noticed oral lesions during the pandemic. We are unfortunately, unable to make this distinction on the types of oral ulcers we collected. We did not this limitation in the discussion and emphasised the possibility of the lesion being the stressed induced aphthous ulcer.

For this reason, we will change the details on the manuscript to self-reported oral ulcers to help with the distinction

We hope this response helps

Morenike Folayan
For the authors